# Intestinal Dysbiosis: Microbial Imbalance Impacts on Colorectal Cancer Initiation, Progression and Disease Mitigation

**DOI:** 10.3390/biomedicines12040740

**Published:** 2024-03-26

**Authors:** Mary Garvey

**Affiliations:** 1Department of Life Science, Atlantic Technological University, F91 YW50 Sligo, Ireland; mary.garvey@atu.ie; 2Centre for Precision Engineering, Materials and Manufacturing Research (PEM), Atlantic Technological University, F91 YW50 Sligo, Ireland

**Keywords:** dysbiosis, inflammation, DNA damage, colorectal cancer, AMPs

## Abstract

The human gastrointestinal tract houses a diverse range of microbial species that play an integral part in many biological functions. Several preclinical studies using germ-free mice models have demonstrated that the gut microbiome profoundly influences carcinogenesis and progression. Colorectal cancer appears to be associated with microbial dysbiosis involving certain bacterial species, including *F. nucleatum*, pks+ *E. coli*, and *B. fragilis*, with virome commensals also disrupted in patients. A dysbiosis toward these pro-carcinogenic species increases significantly in CRC patients, with reduced numbers of the preventative species *Clostridium butyicum*, *Roseburia*, and *Bifidobacterium* evident. There is also a correlation between *Clostridium* infection and CRC. *F. nucleatum*, in particular, is strongly associated with CRC where it is associated with therapeutic resistance and poor outcomes in patients. The carcinogenic mode of action of pathogenic bacteria in CRC is a result of genotoxicity, epigenetic alterations, ROS generation, and pro-inflammatory activity. The aim of this review is to discuss the microbial species and their impact on colorectal cancer in terms of disease initiation, progression, and metastasis. The potential of anticancer peptides as anticancer agents or adjuvants is also discussed, as novel treatment options are required to combat the high levels of resistance to current pharmaceutical options.

## 1. Introduction

Cancer is defined as uncontrolled cell proliferation leading to the formation of neoplasms, which can spread systemically via the blood and lymphatic circulation, termed metastasis. Colorectal cancer (CRC), cancer of the large intestine and/or rectum, is a major contributor to morbidity and mortality globally, as the second deadliest cancer (after lung cancer) in both sexes, with an increased risk of mortality in younger patients <50 years [1]. Colorectal adenocarcinoma accounts for ca. 90% of CRCs, followed by mucinous colorectal adenocarcinoma, medullary CRC, and, rarely, signet ring cell CRC [2]. Colorectal carcinoma results from genetic mutations leading to the transformation of normal intestinal epithelia cells into precancerous lesions (adenomatous intermediate), followed by invasive carcinoma (adenocarcinoma) with potential metastasis to secondary organs, most commonly the liver [2]. The World Health Organisation (WHO) reported 1.9 million cases of CRC in 2020, with 930,000 deaths globally, with Europe (EU), Australia, and New Zealand having the highest incidence rates and Eastern EU having the highest mortality rates [3]. Importantly, 10% of diagnosed cancers and cancer fatalities in 2020 globally were due to CRC [4]. By 2040, the incidence of CRC is predicted to be approximately 3.2 million cases annually, with 1.6 million deaths [5]. The incidence of CRC is higher in males, with rectal cancer being more difficult to treat as it is often associated with tissue invasion, metastasis, recurrence, and additional complications [6]. CRC also has a greater prevalence in persons over 50 from countries with a higher economic status; however, prevalence is increasing in persons <50 years of age in low-income countries [7]. The risk factors for CRC are both genetic and environmental in etiology, and they include age, alcohol and tobacco consumption, a high processed meat diet, a low fruit and vegetable diet, a physically inactive lifestyle, inflammatory bowel disease (IBD), i.e., Crohn’s or ulcerative colitis (UC), familial history, and genetic syndromes such as familial adenomatous polyposis (FAP) and Lynch syndrome [8]. Approximately 20% of CRC cases result from genetic (congenital mutation) or familial predisposition, with an increasing risk of CRC at a younger age [7]. FAP results from a mutation in the APC tumor suppressor gene, which, when accompanied by an accumulation of mutations, leads to the growth of colorectal polyps and CRC [9]. Polyps are growths from the lumen of the colon or rectum and are categorized as hyperplastic (benign) and precancerous adenoma or adenomatous polyps (subdivided into tubular, tubulovillous, and villous adenomas). Intestinal polyps are present in ca. 95% of CRC patients, indicating their significance as precursors to lesions [10]. Widespread screening, early detection, and removal of adenomas have reduced the mortality rates of CRC [11]. The treatment protocols for CRC include traditional approaches, i.e., surgery, radiation, and chemotherapy, with limited long-term efficacy. The systemic treatment options consist of chemotherapy, targeted therapy (monoclonal antibodies, Bevacizumab, and Cetuximab), and immunotherapy [7]. Pharmacodynamic and pharmacokinetic factors, including intestinal pH, drug solubility, non-selective targeting, and adverse drug reactions, impact chemotherapy treatment [7]. Chemotherapeutic drug candidates for CRC, including 5-fluorouracil, oxaliplatin, capecitabine, and irinotecan, are also cytotoxic to normal cells [12]. The classical drug fluorouracil has high resistance, with a response rate of less than 10% in CRC patients [13]. Due to the high prevalence of CRC, high mortality, and treatment resistance, novel approaches to prevent and treat carcinoma of the colon and rectum are therefore urgently needed. To lower incidence and associated mortality, studies are warranted to better establish CRC etiology and disease progression. The association between the gastrointestinal microbiota, IBD, chronic inflammation, and CRC has become an area of much investigation. Alteration in the resident microbial species of the large intestine appears to greatly impact diseases of the gastrointestinal tract (GIT) [14]. This review aims to outline the relationship between GIT dysbiosis, inflammation, and the development of CRC, where insights into novel treatment approaches are also described. Insights into the epigenetic alterations induced by the GIT microbiota and carcinogenesis enable an enhanced understanding of the environmental causes of CRC.

## 2. Intestinal Microbiota

The human microbiota is a diverse range of microbial species, including bacteria, fungi, viruses, phages, and archaea (*M. smithii*), which are predominately located in the GIT [14]. Studies indicate six bacterial phyla present in the GIT, namely, *Firmicutes*, *Bacteroidetes*, *Actinobacteria*, *Proteobacteria*, *Fusobacteria*, and *Verrucomicrobia*, with a dominance amongst the *Firmicutes* and *Bacteroidetes* [15]. The fungal species present include *Candida*, *Saccharomyces*, *Malassezia*, and *Cladosporium* [15]. This mycobiome is much less diverse and abundant than the bacterial colonizing species present, and as such, it is often neglected in terms of disease manifestation [16]. These commensal microbial species and their microbiomes (genetic material) impact human health, disease status, and wellbeing. More recently, the intestinal microbiota has been recognized as an organ due to its role in maintaining body homeostasis, metabolism, intestinal barrier integrity, inflammation, endocrine, and neuro and immune stimulation [17]. A healthy intestinal barrier prevents movement of material into the bloodstream from the intestinal compartment and consists of tight junctions, a mucosal layer, and antimicrobial peptides (AMPs), e.g., defensins, cytokines, and Immunoglobulin A (IgA) [18]. The biosynthesis of essential biological molecules, including vitamins, small-chain fatty acids (SCFAs), amino acids, and lipids, is also performed by the GIT microbiota [19]. The metabolites produced by the resident microbiota have important functions in maintaining local and systemic health (Table 1). Studies show that an imbalance in the microbiota (dysbiosis) is associated with obesity, IBD, neurological disease, cardiovascular disease (CVD), chronic kidney disease, and liver disease, including cirrhosis and carcinoma [20]. Importantly, research using mice and human studies has established the role of the microbiota in the gut–brain axis via secretion of neuroactive biologics, e.g., serotonin, dopamine, acetylcholine, and γ-aminobutyric acid (GABA) [14], influencing neurophysiology, cognitive ability, and behavior [15]. *Escherichia coli*, for example, is a Gram-negative, facultative anaerobe and a part of the intestinal microbiota, where it functions to synthesize vitamin K and sequester oxygen, allowing for the growth of anaerobic species such as the Bacteroidetes. Of the four phylotypes of *E. coli*, type A is commensal, with pathogenicity associated with types B and D [21]. Excellent reviews on the microbial species present, metabolites, and causation of dysbiosis are provided elsewhere [14,17,22]. Intestinal dysbiosis allows for the colonization and proliferation of oncogenic bacteria, which are associated with CRC [23].

### 2.1. Epigenetic Activity of Intestinal Microbiota and CRC

Alterations in the gut microbiota have been identified in CRC and hepatocellular carcinoma patients [26]. Research supports the association between microbial dysbiosis and the formation of cancer, particularly of the digestive tract, stomach, and large intestine. Carcinogenesis is a multistep process consisting of three stages, initiation, promotion, and progression, ultimately leading to uncontrolled cell proliferation, neoplasm, and malignancy [27]. This results from the accumulation of genetic mutations, which can be somatic/acquired or inherited over a number of years with environmental factors, age, diet, and other risk factors associated with an increased prevalence of CRC [1]. The consumption of processed meat, red meat, alcohol, obesity, smoking, and inactivity are associated with incidence of CRC, with a reduced risk associated with high-fiber diets with calcium and vitamin D [28]. The conversion of benign hyperplastic cells to malignant cells is essential for carcinogenesis, with invasion and metastasis resulting from additional genetic and epigenetic changes [27]. Carcinogenesis of the GIT via dysbiosis is associated with mutagenic metabolites, initiation, cell proliferation, chronic inflammation, and malignancy [29]. Importantly, pathogenic species may contribute to tumorigenesis in 20% of cases, with dysbiosis associated with many local and systemic malignancies, primarily of the liver [30]. Several preclinical studies using germ-free mice models have demonstrated that the gut microbiome profoundly influences carcinogenesis and progression. Additional genetic and epigenetic mechanisms are also associated with colorectal cancer [29]. Particularly, dysbiosis involving major bacterial species *Enterococcus faecalis*, *Escherichia coli*, *Bacteroides fragilis*, *S. bovis*, *Fusobacterium nucleatum*, and *Heliobacter pylori* is believed to be associated with the development of CRC [15]. *H. pylori* is long established as a cancer-causing pathogen (stomach cancer) and classified as a group 1 carcinogen by the International Agency for Research on Cancer (IARC) [31]. A dysbiosis toward pro-carcinogenic species (*B. fragilis*, *Enterococcus faecalis*, *E. coli*, *F. nucleatum*, *Peptostreptococcus anaerobius*, *Porphyromonas*, and *Micromonas parvum*) increases significantly in CRC patients, with reduced numbers of the preventative species *Clostridium butyicum*, *Roseburia*, and *Bifidobacterium* evident [25]. There is a correlation between *Clostridium* infection and CRC in ca. 40% of patients, manifesting in both when the acidic and hypoxic environment produced by CRC encourages *Clostridium* germination and proliferation [32]. Fungal dysbiosis in CRC patients is associated with an increase in *Microbotryomycetes*, *Sordariomycetes*, *Microascaceae*, *Sordariales*, *Lasiosphaeriaceae*, and *Microascales* species [16]. Research has confirmed the relationship between fungal dysbiosis and IBD, including a decreased prevalence of *S. cerevisiae* and an increased presence of *Candida albicans* [18]. Importantly, the prevalence of colitis-associated CRC has decreased in recent years due to improved therapeutic options and increased surveillance for CRC in IBD patients [33].

### 2.2. Dysbiosis Initiates Colorectal Cancer

Colonization of the GIT with pathogenic species or with an overabundance of non-beneficial species is associated with the production and secretion of microbial toxins (Table 2). Currently, there are three known genotoxins produced by bacterial species that target DNA, resulting in strand breaks: cytolethal distending toxin (CDT) produced by Gram-negative bacteria, typhoid toxin produced by *Salmonella enterica* serovar *Typhi* (*S. Typhi*), and thirdly, colibactin produced by group B2 *E. coli* [34]. *B. fragilis* produces and secretes endotoxins, namely the *B. fragilis* toxin, that can cause DNA damage, leading to mutations and colon cancer initiation [35]. Strains excreting the *B. fragilis* toxin have promoted CRC tumorigenesis in mouse models via DNA damage and cell proliferation of epithelial cells [36]. Genomic instability results if such strands break or genetic mutations are not repaired, which may result in tumor initiation and progression [30]. Additionally, certain pathogenic species, such as *Shigella flexneri*, inhibit DNA repair pathways by enzymatic degradation of p53 proteins, increasing the risk of mutations escaping repair [30]. P53 is a tumor suppressor gene that imitates cell cycle arrest and apoptosis in DNA damaged cells, where a mutation or suppression of this gene or protein leads to uncontrolled cell proliferation [15]. Like most cancers, CRC has a high prevalence of p53 mutations. Toxins produced by fungal species may also contribute to carcinogenesis, including aflatoxin B1 and patulin produced by *Aspergillus* species, which cause the formation of DNA adducts and DNA strand breaks, and reactive oxygen species (ROS), respectively [16]. ROS cause CRC initiation via inflammation, DNA damage, epithelial-to-mesenchymal transition, apoptosis, and angiogenesis [37]. ROS also result in the initiation of mutations in various proteins involved with regulating the cell cycle (proto-oncogenes), including p53, Ras, and c-Myc, leading to oncogene activation [12]. Colibactin produced by enterotoxigenic *E. coli* with polyketide synthesis (pks) genomic islands alters cell cycle progression and induces DNA damage [21]. Such pks+ *E. coli* has promoted CRC tumorigenesis in experimental mouse models [36]. Pathogenic bacteria can also bind to intestinal epithelial cells and promote cell growth and proliferation, leading to hyperplasia.

### 2.3. Dysbiosis Causes Inflammatory Carcinogenesis

Approximately 80% of CRC cases are sporadic and present in persons without genetic or familial predispositions [7]. Sporadic CRC (not associated with familial or inherited factors) is associated with chronic IBD, particularly UC, inactive lifestyles, poor diet, and alcohol [38]. Chronic inflammation, as present in IBD patients, is associated with incidences of CRC [33]. Crohn’s disease of the ileocolic region, for example, may increase CRC risk. Importantly, IBD patients have a 60% increased risk of developing CRC compared to non-IBD patients [38]. Enterotoxigenic (ETBF) strains of *B. fragilis* are associated with CRC and colitis [9]. Interestingly, *B. fragilis* is the most frequently isolated opportunistic anaerobic pathogen in clinical cases of diarrhea, sepsis, and extra-intestinal infections [39]. Chronic infection, pathogenic virulence factors, and associated inflammation promote oncogenic activity in local cells [31]. An overabundance of pathogenic fungi, including *Malassezia restricta*, is associated with Crohn’s disease and IBD [40]. *C. albicans* is associated with inflammation due to the production of the cytosolic peptide toxin (candidalysin), which is cytotoxic to epithelial cells and promotes inflammation [18]. The migration of immune cells and the inflammatory response lead to inflammation-driven carcinogenesis due to the production of ROS, which damage DNA, causing double-strand breaks [37]. Innate immunity is supported by the interaction between the resident microbiota and specific receptors termed pathogen recognition receptors (PPRs) on the surface of immune cells, which are activated by microbial pathogen-associated molecular patterns (PAMPS) [41]. PPRs include Toll-like receptors (TLRs) and Nod-like receptors (NLRs), amongst other types, which trigger a cascade of signals, leading to the production of pro-inflammatory cytokines (IL1, IL 18, and IL 6), chemokines, and growth factors involved in intestinal cell integrity and repair. Alterations in these PPRs, such as TLRs, can result in excessive inflammatory responses, inflammasome activation, and alterations in dysbiosis, which are associated with cancer progression. Studies have demonstrated the relationship between inflammation-associated CRC in TLR-deficient mice models and increased size of tumors and altered cytokine production [41].

The lipopolysaccharide (LPS) component of the Gram-negative cell wall is known to activate the inflammasome [42], which has a cascade effect on cytokines and interleukins, converting them to active pro-inflammatory agents, resulting in inflammation [29,43]. Inflammasomes recognize PAMPs and act as links between the microbiota and the host immune system. Studies suggest the importance of inflammasome activity and the inflammasome–microbiota axis in several disease states, including CRC [44]. LPS is associated with cell adhesion, cell degradation, and cell invasion in CRC metastasis [43]. Studies have demonstrated that LPS-associated inflammation increases the activity of oncogenic genes and the cell proliferation of CRC in IBD animal models [12]. LPS also alters the integrity of the intestinal barrier by disrupting the tight junctions, leading to CRC tumorigenesis, tumor growth, leakage of microbial products into the bloodstream, systemic inflammation, and metabolic endotoxemia (ME) [43]. The LPS toxins of *E. coli* species have greater immunogenic potential than the LPS excreted by other Enterobacteria. A dysbiosis involving *Fusobacterium*, *Clostridium*, *Prevotella*, *Desulfovibrio*, and *Enterococcus* can lead to the formation of inflammation metabolites, including elevated levels of trimethylamine N-oxide (TMAO) and decreased levels of SCFAs [26]. SCFAs help maintain intestinal integrity and are associated with increased anticancer activity, including apoptosis in CRC cells [45]. SCFAs also have functions in immune regulation, anti-inflammatory effects, and the regulation of the host microbiota. The gut microbiota regulates bile acid metabolism and biosynthesis with alterations in bile acid concentrations associated with intestinal inflammation, impaired Farnesiod x receptor function, and CRC [46]. Commensal *Bifidobacterium* decreases intestinal pH, inhibits the growth of pathogenic *E. coli*, and inhibits the virulence genes of pathogenic species [47]. Mucosal biofilm formation is associated with CRC in sporadic patients and may influence the progression of familial CRC [9]. Indeed, mucosal invasive biofilms have been detected in ca. 50% of CRC patients, compared to 13% of healthy patients [48]. Biofilms, which are a community of microbial cells present in a matrix of extracellular polymeric substances (EPS), are present in the intestines of healthy and non-healthy persons. The biofilms present in CRC patients contain *Veillonellaceae*, *Lachnospiraceae*, *Coriobacteriaceae*, *Bacteroidetes*, and *Firmicutes* species, CRC developed in mice transplanted with samples from CRC patients [48]. The presence of biofilms promotes inflammatory interleukin 6 (IL-6) activity and signal transducer and activator of transcription (STAT) signaling, which supports tumor proliferation and carcinogenesis and damages tight junctions, allowing for tumor invasion and metastasis [21]. Research has demonstrated the presence of pks+ *E. coli* and *B. fragilis* toxin-producing species in the mucosal biofilms of FAP patients [36]. CRC patients’ biofilms tend to be polymicrobial, containing *B. fragilis*, *E. coli*, and *F. nucleatum* [49]. *F. nucleatum* (and CRC-relevant subspecies: *vincentii*, *animalis*, and *polymorphum*) is a Gram-negative bacillus that promotes cell proliferation and tumorigenesis via virulence (expressing FadA and Fap2 adhesins)-induced inflammation and proinflammatory cytokines, IL 6 and IL 8, and pro-metastatic cytokine activation [50]. *F. nucleatum* LPS induces resistance to anticancer therapeutics in CRC cells and is associated with more severe outcomes and mortality in patients [51]. Studies have shown that *F. nucleatum* causes the progression of CRC in mouse models [52]. Studies show that *F. nucleatum*-activated autophagy of CRC cells via TLR pathways results in resistance to oxaliplatin and fluorouracil [53,54]. While the microbiota–autophagy axis appears important in cancer progression, more studies are warranted on its specific molecular mechanisms to improve anticancer treatment in clinical settings.

Microbial biofilms are antibiotic-resistant and host-immunity-resistant, resulting in increased inflammatory reactions and cytokine excretion with resident species, such as those described, that possibly produce bacterial-derived genotoxic compounds. Research investigating the presence of microbiota species in CRC patients and adenomatous polyps offers evidence of the presence of certain pathogens in the tumor environment [53]. The question remains, however, of causation and correlation; are such pathogens drivers of carcinogenicity and metastasis, or are they migrating toward a more favorable environment?

## 3. Colorectal Cancer Treatment

The treatment of CRC involves traditional approaches such as surgery, chemotherapy, using anticancer active pharmaceutical ingredients (APIs), and radiation therapy, with these strategies failing to eradicate the disease completely. Oxaliplatin and fluorouracil are the APIs of choice for CRC treatment; oxaliplatin, however, is associated with peripheral neuropathy, and fluorouracil is associated with GIT and liver adverse drug reactions [54]. Pharmacological issues, including drug solubility, drug resistance, lack of target specificity, and associated adverse side effects, are issues with the current therapeutic options. Cancer cells develop resistance to chemotherapeutics via gene mutations, metabolic alterations, and epigenetic modifications [55]. Therapeutics are applied as local, systemic, or combined treatments, depending on the stage of tumor progression. Research has moved toward novel therapeutic approaches, including immunotherapy and nanomaterial carrier systems as anticancer agents such as polymeric nanoparticles, graphene oxide, liposomes, metal oxides, and inorganic nanoparticles, amongst others [7]. Additional novel treatment options showing promise include immune checkpoint inhibitors (ICIs), chimeric antigen receptor (CAR) T cell therapy, T cell receptor (TCR) alterations, RNA-based therapies, and cytokine therapy [56]. More recently, there has been growing interest in the application of fecal matter transplants (FMTs) and gut microbiota modification (pro and prebiotics) to restore a healthy microbiota in chronic cases of disease, including IBD and cancer [57]. FMTs possess some risks due to the possible transmission of pathogenic species and antibiotic-resistant bacteria to the patient [58].

### 3.1. Modification of Host Microbiota against CRC

As research provides increasing insight into the role of the GIT microbiota in the initiation and progression of CRC, restoring a healthy gut microbiota may offer a prophylactic strategy to prevent or treat this problematic disease. Dietary alterations promote the proliferation of beneficial microbial colonizers and the death of pathogenic species (*Fusobacterium*, *Clostridium*, *pks*+ *E. coli*, *B. fragilis*, and *Enterococcus*) associated with increased inflammation, and CRC may reduce the risk of CRC and improve prognosis in patients [57]. Lactic acid bacteria and the *Lactobacillus* species are the most commonly studied probiotic species. Studies describe the use of probiotics containing *L. rhamnosus* GG to modify the gut microbiota with an increased expression of anticancer agents, including P53, caspase-3, and interleukin 2, which encourages anticancer immune activity and the downregulation of pro-cancer mediators [59]. The administration of the *L. casei* BL23 strain to a mouse model downregulated IL-22 (a tumor promoting cytokine) and upregulated anticancer caspase-7 and caspase-9 [60]. *L. reuteri* produces an antimicrobial agent reuterin, which has potent anticancer activity in mice models [61]. Reuterin reduces CRC cell proliferation and survival, restricting tumor growth in vivo [61]. Studies also describe the anti-inflammatory effects of *Lactobacillus* species in IBD mice models, reducing the expression of inflammatory genes such as STAT3 [62]. STAT3 signaling is active in IBD and CRC, promoting inflammation, tumorigenesis, and metastasis [21]. *L. paracasei* was found to induce CRC cell apoptosis and inhibit proliferation by regulating the expression of the specific Bcl-2 family of apoptosis proteins [63]. *Lactobacillus* appears to prevent inflammation and DNA damage via the antioxidant glutathione and restore tight junctions [64]. Promoting the production of SCFAs may inhibit carcinogenesis by suppressing cell growth, migration, and tissue invasion, as demonstrated in mice models [65]. Butyrate is protective against xenobiotic-induced DNA damage and initiates apoptosis via cell signaling mechanisms [66]. Studies have assessed the levels of SCFAs in CRC patient’s compared to healthy controls with variable findings, but no conclusive evidence exists on the SCFA profile of CRC patients [67]. The findings of Alvandi et al. (2022), however, concluded that intestinal SCFA levels are associated with CRC risk and progression and may act as biomarkers or as drug therapy in CRC management [67]. When SCFA butyrate was used in combination with the anticancer drug Oxaliplatin, inhibition of cell proliferation, invasion, and metastasis was observed with increased apoptosis in CRC cells [68]. Studies demonstrate that butyrate significantly improves the clinical symptoms of Crohn’s disease in patients [69]. Butyrate has demonstrated efficacy in preclinical trials and clinical studies for the treatment of IBD [70]. A clinical trial (NCT05218850) aims to investigate the efficacy of therapeutic butyrate against UC in 7- to 21-year-old patients. *Clostridium butyricum*, *L. plantarum*, and *Butyricicoccus pullicaecorum* are butyrate-producing species associated with anticancer action in CRC models [71]. Studies have investigated the use of the probiotic *Propionibacterium freudenreichii* due to its production of acetate and propionate SCFAs, which prevent CRC cell proliferation and induce cell cycle arrest in CRC cells [13]. The research of Yu et al. (2023) reversed the dysbiosis in CRC mice models and alleviated disease progression via anti-inflammatory and anticancer immune activity [32]. Further studies are warranted, however, as high doses of prebiotics may have negative effects on glucose metabolism where probiotics are not suitable for immune-compromised patients [58].

### 3.2. Antimicrobial Peptides against CRC

Antimicrobial peptides (AMPs) show potential as anticancer agents or anticancer peptides, with a multi-hit mode of action on cancer cells, incorporating cell membrane disruption, apoptosis induction, and anti-proliferative and anti-inflammatory action [55]. Furthermore, AMPs or their fragments can be engineered to improve efficacy and selectivity [72]. Studies assessing the activity of AMPS Melittin (from Bee venom), Cecropin A (insect AMP), and a Cecropin A/Melittin hybrid against HT-29 and HCT-116 cells in vitro have demonstrated anticancer activity [73]. Similarly, the AMP microcin E492 (a bacteriocin produced by *Klebsiella pneumoniae*) demonstrated cytotoxicity to colorectal cells in vitro [74]. A MELITININ+BMAP27-conjugated peptide provided apoptosis-induced death of CRC cell lines, namely HT29, SW742, and HCT-116, in vitro [75]. The human-derived AMP LL37 of the cathelicidin family and its analogues induced apoptosis in cancer cells by upregulating the apoptosis genes Bax and Bak [72]. The anticancer effects of LL37 against CRC cells were selectively time- and dose-dependent, with no toxicity observed in healthy cells [76]. The AMP Jelleine-I (J-I), from the royal jelly of honeybees, and its derivative Br-J-I do not display potent anticancer activity in CRC models but do have antibacterial activity against *F. nucleorum* and associated CRC development [58]. When Br-J-I was used in conjunction with API 5-FU (5 µM), a 40% toxicity to CRC cells was achieved in vitro, highlighting the adjunctive potential of this AMP [58]. The AMP RT2 from Crocodylus siamensis leukocytes displayed potent anticancer action against human colon cancer (CACO2) cells in vitro via anti-proliferation action at a proteomic level [55].

AMPs also possess anti-biofilm activity, which may aid in eradicating the presence of CRC-associated biofilm species such as *F. nucleorum*, *pks+ E. coli*, and *B. fragilis*. For example, the AMPs AG-30, AG-30/5C, WRL3, melimine, 73c, and D-73 have all demonstrated anti-biofilm action [14]. The AMP lactoferrin (12.5 μg/mL) inhibited the biofilm formation of *B. fragilis* in vitro [77]. Studies are warranted to determine the in vivo efficacy of AMPs against CRC-specific pathogens, their biofilms, and their associated anticancer impact.

AMPs show potential as anticancer agents or anticancer peptides as they target cancer cells more selectively with reduced cytotoxicity to healthy cells [78]. While studies highlight the potential of AMPs in cancer treatment, hurdles exist, limiting their application in a clinical setting. Namely, issues relating to large-scale production, formulation, and pharmacokinetic and pharmacodynamic limitations must be overcome to successfully implement AMPs as anticancer peptides (Table 3). Additionally, a full biocompatibility and toxicological risk assessment must be made in accordance with the regulatory guidelines to ensure patient safety.

### 3.3. Intestinal Virome and CRC

Similarly to AMPs, bacteriophage (phage) and phage-derived endolysins may offer antibacterial action against CRC pathogens. The application of bacteriophages in the control of AMR infectious disease shows the potential of these selective and potent agents for clinical use. Furthermore, phages have anti-biofilm activity, and phage cocktails are more effective at preventing and eradicating biofilms in vitro [80]. For example, the phage vB_BfrS_23 infects and kills certain *B. fragilis* strains, with ϕ B124-14 active against 5 out of 15 tested *B. fragilis* spp. [81]. Studies have isolated and described five phages that are active against *F. nucleatum* and concluded that phage JD-Fnp4 may have significant potential for clinical application [82]. Human GIT commensal phages, the CrAssphages, are believed to infect *Bacteroidales* bacteria [83]. Dysbiosis of the GIT virome is associated with IBD and *C. difficile* infection [84], with substantial amounts of the bacteriophage microbiome absent or lessened in IBD patients, which may contribute to dysbiosis of bacterial commensals and eubiosis [85]. Furthermore, studies have altered the gut virome of mice with CRC-induced neoplasia [86]. Studies have investigated the use of phages against CRC-associated pathogens as biomarkers for CRC diagnosis [87]. Phages are thought to be key players in bacterial-associated carcinogenesis by modulating the microbial diversity present and CRC development [23]. One study identified four phages in increased abundance in CRC patients [88]. Cancer virotherapy is immunotherapy that uses genetically modified viruses (oncolytic viruses) to selectively target and lyse cancer cells and also facilitates the activity of immune checkpoint inhibitors, which may be beneficial in the treatment of CRC [89]. The identification of phages’ activity against oncogenic bacteria in the tumor microenvironment may offer a targeted therapeutic approach. Phages may be genetically engineered to expand their host range against different bacterial species and to carry anticancer AMPs, cytokines, or antibodies [40]. The studies of Asavarut et al. (2022) designed a cytokine gene delivery system based on recombinant adeno viral DNA and bacteriophage coat proteins with anti-tumor activity in vivo [90].

## 4. Conclusions

Colorectal cancer represents a difficult-to-treat cancer with increasing prevalence globally. Sporadic cases of disease are associated with diet and lifestyle choices that are typically connected with Western societies. Inflammation is also a key driver of CRC, with IBD patients having a 60% increased risk of developing CRC compared to non-IBD patients. Colorectal cancer appears to be associated with microbial dysbiosis involving select bacterial species, namely, *F. nucleatum*, *pks+ E. coli*, and *B. fragilis*, with virome commensals also disrupted in patients. *F. nucleatum*, in particular, is strongly associated with CRC, where it is associated with therapeutic resistance and poor outcomes in patients. The carcinogenic mode of action of pathogenic bacteria in CRC is a result of genotoxicity, epigenetic alterations, ROS generation, and pro-inflammatory activity. Microbial dysbiosis is, therefore, a risk factor for CRC development and progression. Certain species of phages and *F. nucleatum* may potentially offer a diagnostic tool for CRC as biomarkers, as the microbiota and microbiome differ in CRC patients compared to healthy persons. Modifying the microbiota may aid in disease treatment and prevention. SCFA-producing species, for example, are key players in immune homeostasis, anti-inflammatory activity, and the maintenance of intestinal tight junctions. SCFAs are also active in tumor suppression, cell cycle arrest, and apoptosis in CRC cells. The alteration of the gut microbiota to increase SCFA levels in CRC patients may aid in disease treatment. The use of SCFA-producing species as probiotics may act as adjuvant therapeutic strategies in the treatment of CRC. Certain AMPs demonstrate anticancer activity in vitro, with selectivity toward cancer cells over non-cancerous cells. AMP cytotoxicity is associated with membrane damage and the induction of apoptosis. AMPs also possess potent antimicrobial action, including anti-biofilm activity. Colon mucosal biofilms housing pathogenic species are associated with carcinogenesis in CRC patients. AMPs may offer a protective role by eradicating such biofilms and limiting CRC progression. Research is warranted investigating the potential of AMPS and bacteriophages as adjuvant treatments for CRC, incorporating biocompatibility, formulation, drug delivery, and efficacy studies. The application of bioprocessing, DNA technology, and genetic engineering may enable large-scale production and the development of more potent and safe anticancer peptides. Additionally, investigative studies are warranted to determine the relationship between the microbiota–autophagy axis and the inflammasome–microbiota axis to improve therapeutic success and patient prognosis.

## Figures and Tables

**Table 1 biomedicines-12-00740-t001:** GIT commensal species, metabolites produced, and dysbiosis-associated morbidities.

Microbiota Species	Metabolites Produced	Function	Diseases Associated with Dysbiosis
*Blauia*, *Coprococcus* and *Roseburia* species [22]	Short-chain fatty acids, butyrate, and propionate [14].	Regulate diet, insulin, weight, role in gut–brain connection, and neurotransmitter activity [14].	Neurological issues, major depression, Autism, Parkinsons disease [14], liver, heart, and kidney disease, Crohn’s disease, and colorectal cancer [22].
*Fusobacteria*, *Bacteroidetes*, *Fusobacteria*, *Proteo-bacteria*, and *Firmicutes* [24]	Vitamins, e.g., B group vitamins and vitamin K [14].	DNA replication, immunity, and red blood cell formation [17].	Heart failure, neuropathy, and anaemia [24].
Gram-positive bacteria firmicutes (*Lactobacillus* and *Enterococcus*) and certain Gram-negative bacteria Bacteroidetes [24,25]	Bile acids, e.g., cholate, hyocholate, deoxycholate, taurohyocholate, and ursodeoxycholate [22].	Facilitate lipid and vitamin absorption; regulation of gut microbiota composition, hormonal and immune functions [22], and homeostasis of cholesterol.	Cholangitis, atherosclerosis, UC, cancer, hepatic encephalopathy, multiple sclerosis, Alzheimer’s disease, and Parkinson’s disease [14].
*Streptomyces*, *Bacillus*, *Pseudomonas*, *Klebsiella*, and *Staphylococcus* species [14]	Induce immune modulators, i.e., cytokines and interleukins.	Immune modulation and neuro-immune stimulation.	Mood disorders, neurodegenerative disorders, and fibromyalgia [14].
*Lactobacillus plantarum*, *Proteus vulgaris*, *Bacillus*, and *Serratia marcescens*, *Lactobacillus*, *and Bifidobacterium* [14]	Neurotransmitters, e.g., serotonin, dopamine, glutamate, etc. [14].	Enteric nerve stimulation and systemic nerve interaction [14].	Mood disorders and functional somatic syndromes [14].
*Clostridium*, *Bacillus-Lactobacillus-Streptococcus*, *Proteobacteria* (small intestine), *Clostridia*, and *Peptostreptococci* (large intestine) [24]	Branched-chain amino acids (BCAAs) [24]. Amino acid Phenylacetylglutamine [19].	Synthesis substrates, T cell function [24], and agonists of B-adrenergic receptors.	Insulin resistance, cancer [24], and cardiovascular disease [19].

**Table 2 biomedicines-12-00740-t002:** Toxin produced by species associated with intestinal dysbiosis and pro-cancer activity.

Pathogen	Toxin	Activity	Species
Bacterial	Cytolethal distending toxins	DNA strand breaks.	*Escherichia coli*, *Aggregatibacter actinomycetemcomitans*, *Haemophilus ducreyi*, *Shigella dysenteriae*, *Campylobacter* sp., and *Helicobacter* sp.
	Typhoid toxin		*S. Typhi*.
	Colibactin	Causes cell proliferation and depletes CD3+ T cells [29].	*E. coli* strains of the phylogenetic group B2.
	Pasteurella multocida toxin (PMT) [29]	Prevents apoptosis and signaling pathways in carcinogenesis [29].	*Pasteurella multocida* [29].
	*B. fragilis* toxin (fragilysin) [35]	ROS generation, proinflammatory, and biofilm formation [35].	* Bacteroides fragilis. *
	AvrA protein [31]	Promotes colonic epithelial cell proliferation [31].	* Salmonella * species.
	Lipopolysaccharide toxin (LPS)	Inflammasome activator [12].	Gram-negative species.
Fungal	Candidalysin	Causes the release of pro inflammatory mediators and stimulates tissue growth and angiogenesis [16].	*Candida albicans* [16].
	Aflatoxins, e.g., aflatoxin B1 (AFB1) Patulin toxin	Formation of DNA adducts, DNA strand breakage, oxidative damage, and ROS generation [16].	* Aspergillus * species.

**Table 3 biomedicines-12-00740-t003:** Advantages and current limitations of AMPs as therapeutic options in the treatment of CRC.

Advantages	Limitations
Appear more toxic to cancer cells, i.e., selectively anticancer [76].	Biocompatibility needs to be established [78].
Anticancer action demonstrated in vitro for many AMPs [55,73].	No clear in vivo efficacy as anticancer agents [55].
May be used in combination therapy with current anticancer APIs [58,72].	Limited stability and short half-life [58].
Easier to synthesize—short amino acid sequences [78].	Protein and enzymatic degradation in vivo mean low oral bioavailability [72].
Also possess immunomodulatory action—anti-inflammatory [78].	Pharmacokinetic and pharmacodynamic profiles need to be established [78].
No immunogenicity [58].	Over stimulation of immune system may be an issue, i.e., cytokine storm [78].
Some effective against biofilms—pathogenic species associated with CRC [77].	Expensive production costs [78].
Short peptides can be engineered to improve efficacy and delivery [72].	Large-scale production issues need to be overcome, e.g., fermentation considerations and formulation considerations [78].
Some AMPs are stable and active in a wide pH range [78].	Post translational modifications and downstream isolation of AMPs may hinder production [79].

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
