# Peer review of "Intestinal Dysbiosis: Microbial Imbalance Impacts on Colorectal Cancer Initiation, Progression and Disease Mitigation"

_biomedicines, 2024, doi:10.3390/biomedicines12040740_

Round 1

Reviewer 1 Report

Comments and Suggestions for Authors

The review is well written, but I have some advice for improving the quality. The author concentrates on microbiota in CC. However, the gut microbiome characteristics in individuals with cancer are not characterized quite well in the review. The table and additional section about studies on gut microbiota characteristics in patients are needed. 

The author describes clinical implications and therapeutic strategies to improve gut microbiome in CC. Also, adding the section about probiotics/probiotic therapy in individuals in CC is recommended.

Some less important advice:

1. The background in the abstract section is long. The conclusions/results of the review should be more highlighted.

The author often uses "alarmingly" in the manuscript. This emotionally charged phrase does not fit in a scientific article.

Line 200: enterobacteria - please verify letter case 

Line 151: verify space

There are references in some rows in Table 1, and there are no references in others. Please add references in all rows or standardize it in another way.

Reviewer 2 Report

Comments and Suggestions for Authors

Colorectal cancer is a serious threat to human health worldwide. It is necessary to study the pathogenesis and preventive measures of colorectal, which is a subject of great interest to researchers. Because of this, there are many reviews on intestinal microbiota and colorectal cancer. Compared with the existing reports, the authors' review of the relationship between antimicrobial peptides, intestinal virome and CRC is a novel entry point. However, the microbial species and their impact on colorectal cancer are not sufficiently discussed in this paper. In addition, the logic of this review is confusing, especially in the introduction part.

Reviewer 3 Report

Comments and Suggestions for Authors

The role of intestinal dysbiosis in CRC pathogenesis is an important topic. The author of the article describes the relationship between dysbiosis and CRC initiation and progression quite well, and she is well aware of the therapeutic aspects based on modulating this relationship. 
However, the manuscript does not touch upon aspects of the microbiome that are important both in the pathological role of the microbiome and in the development of cancer: it does not discuss the TLR receptor sensing of innate immunity, and it does not explore the importance and therapeutic potential of the microbiome-inflammasome-CRC and microbiome-TLR-autophagy-CRC axes.
The presentation of these topics is critical to accurately summarizing the article's main message.
A major revision is necessary. 

Round 2

Reviewer 1 Report

Comments and Suggestions for Authors

The authors updated the manuscript quite well.

I have no other concerns.

One suggestion: [53][54] should be written after coma

Reviewer 2 Report

Comments and Suggestions for Authors

The author's revised manuscript has been significantly improved and I recommend it to be accepted for publication.

Reviewer 3 Report

Comments and Suggestions for Authors

The revised version of the manuscript iscorrect. Now, I find it acceptable for publication.